# Individualized Perioperative Hemodynamic Management Using Hypotension Prediction Index Software and the Dynamics of Troponin and NTproBNP Concentration Changes in Patients Undergoing Oncological Abdominal Surgery

**DOI:** 10.3390/jpm14020211

**Published:** 2024-02-16

**Authors:** Jolanta Cylwik, Małgorzata Celińska-Spodar, Mariusz Dudzic

**Affiliations:** 1Anesthesiology and Intensive Care Unit, Mazovia Regional Hospital, 08-110 Siedlce, Poland; 2Anesthesiology and Intensive Care Unit, The National Institute of Cardiology, 04-628 Warsaw, Poland; 3Critical Care, Edwards Lifesciences, 00-807 Warsaw, Poland

**Keywords:** hypotension prediction index, NTproBNP, biomarkers, cardiovascular complications, PMI, AKI, MACE

## Abstract

Introduction: Abdominal oncologic surgeries pose significant risks due to the complexity of the surgery and patients’ often weakened health, multiple comorbidities, and increased perioperative hazards. Hypotension is a major risk factor for perioperative cardiovascular complications, necessitating individualized management in modern anesthesiology. Aim: This study aimed to determine the dynamics of changes in troponin and NTproBNP levels during the first two postoperative days in patients undergoing major cancer abdominal surgery with advanced hemodynamic monitoring including The Acumen^TM^ Hypotension Prediction Index software (HPI) (Edwards Lifesciences, Irvine, CA, USA) and their association with the occurrence of postoperative cardiovascular complications. Methods: A prospective study was conducted, including 50 patients scheduled for abdominal cancer surgery who, due to the overall risk of perioperative complications (ASA class 3 or 4), were monitored using the HPI software. Hypotension was qualified as at least one ≥ 1 min episode of a MAP < 65 mm Hg. Preoperatively and 24 and 48 h after the procedure, the levels of NTproBNP and troponin were measured, and an ECG was performed. Results: We analyzed data from 46 patients and found that 82% experienced at least one episode of low blood pressure (MAP < 65 mmHg). However, the quality indices of hypotension were low, with a median time-weighted average MAP < 65 mmHg of 0.085 (0.03–0.19) mmHg and a median of 2 (2–1.17) minutes spent below MAP < 65 mmHg. Although the incidence of perioperative myocardial injury was 10%, there was no evidence to suggest a relationship with hypotension. Acute kidney injury was seen in 23.9% of patients, and it was significantly associated with a number of episodes of MAP < 50 mmHg. Levels of NTproBNP were significantly higher on the first postoperative day compared to preoperative values (285.8 [IQR: 679.8] vs. 183.9 [IQR: 428.1] pg/mL, *p* < 0.001). However, they decreased on the second day (276.65 [IQR: 609.4] pg/mL, *p* = 0.154). The dynamics of NTproBNP were similar for patients with and without heart failure, although those with heart failure had significantly higher preoperative concentrations (435.9 [IQR: 711.15] vs. 87 [IQR: 232.2] pg/mL, *p* < 0.001). Patients undergoing laparoscopic surgery showed a statistically significant increase in NTproBNP. Conclusions: This study suggests that advanced HPI monitoring in abdominal cancer surgery effectively minimizes intraoperative hypotension with no significant NTproBNP or troponin perioperative dynamics, irrespective of preoperative heart failure.

## 1. Introduction

It is estimated that approximately 300 million surgical procedures are conducted each year globally [1], of which noncardiac surgeries (NCSs) comprise roughly 85% [2]. A significant subset of these surgical interventions is dedicated to treating patients with gastrointestinal malignancies. This is because, notwithstanding the progress in chemotherapy and radiotherapy, surgical intervention continues to be a crucial component in the therapeutic regimen for these conditions.

Despite the introduction of increasingly advanced surgical techniques and a trend toward minimizing invasiveness (e.g., laparoscopy) as well as refining the precision of surgical methods (robotic techniques), patients remain susceptible to perioperative complications associated with the undergone procedure, of which cardiovascular complications are the most common [3,4,5].

In the cohort study published by Spence et al. [6] in 2019, which included 40,000 patients, it was demonstrated that in the group of patients over 45 years old undergoing NCS, serious cardiac and cerebrovascular complications occurred within 30 days of the procedure in one out of every seven patients—indicating a substantial scale of the problem. Moreover, the occurrence of major cardiovascular events (MACEs) increases the risk of mortality within 30 days post-procedure [7,8,9], and the 30-day mortality rate for patients who suffer perioperative myocardial injury (PMI) is approximately 10% [9,10].

Given the volume of surgical procedures, the prevalence of discussed complications, and their serious outcomes, contemporary anesthesiology should prioritize the monitoring and prompt treatment of cardiovascular episodes. The perioperative diagnosis of PMI presents considerable challenges due to the unique characteristics of this timeframe. Symptoms such as chest pain may be obscured by the painkillers given to patients, and conducting imaging tests can be challenging or unfeasible. In line with the European Society of Cardiology (ESC) and American Heart Association (AHA) guidelines, diagnoses should rely on perioperative surveillance. This involves the meticulous observation of changes in routine postoperative electrocardiograms and the measurement of troponin levels, a biomarker indicative of myocardial injury [8,9].

The N-terminal pro B-type Natriuretic Peptide (NTproBNP), an additional cardiac biomarker, has an increasing role in risk stratification [9]. As a critical marker of cardiac function, NTproBNP frequently elevates in response to cardiac stress or dysfunction. Alongside other natriuretic hormones, such as the Atrial Natriuretic Peptide (ANP) and Brain Natriuretic Peptide (BNP), it is synthesized in response to increased ventricular pressure or volume. Collectively, these hormones contribute to regulating fluid balance, natriuresis, and vascular resistance [9,11,12,13], and their levels correlate with the severity and prognosis of heart failure (HF) [12]. Beyond the context of HF, a preoperative elevation in NTproBNP has been associated with an augmented risk of postoperative cardiovascular events [9,14,15,16].

One of the factors most strongly associated with MACE, such as PMI or acute kidney injury (AKI), is intraoperative hypotension (IOH).

One of the methods to individualize the approach to hemodynamic management during anesthesia for elevated risk procedures to reduce the risk of intraoperative hypotension is using, beyond basic blood pressure measurements, advanced hemodynamic parameters such as stroke volume, stroke volume variation, cardiac output, and systemic vascular resistance. These parameters are derived in real-time from the shape and characteristics of the arterial waveform using proprietary algorithms [17]. In addition, applying The Acumen^TM^ Hypotension Prediction Index software (number 02.03.000.103) (HPI) (Edwards Lifesciences, Irvine, CA, USA) for hemodynamic monitoring involves predictive analytics to foresee and prevent episodes of hemodynamic destabilization. It was developed using machine learning to predict arterial hypotension, defined as a mean arterial pressure (MAP) of less than 65 mm Hg for at least 1 min. Additionally, this technology identifies its causes, like hypovolemia, vascular resistance issues, or reduced cardiac contractility. It allows for early, tailored treatment, reducing the frequency and severity of hypotensive episodes. An intervention is recommended when the HPI reaches a threshold of 85 [5,17,18,19,20].

A series of randomized, single-center studies have indicated that the integration of the Hemodynamic Performance Index (HPI) into perioperative monitoring is associated with a reduction in intraoperative hypotension in general noncardiac surgeries. This observation was reported in the HYPE clinical trial conducted by Wijinberge et al. [21], as well as in studies focusing on major noncardiac surgeries involving populations at an elevated risk, as examined by Sribar et al. [22] and Yoshikawa et al. [23]. Complementing these findings, a non-randomized but substantial registry study, the European multi-center prospective observational EU HYPROTECT, conducted by Kouz et al. [24], also demonstrated low incidences of hypotension when using HPI. This study, involving 702 patients across 12 medical centers in five European countries, found that the median time-weighted average mean arterial pressure (MAP) < 65 mm Hg was significantly low, suggesting that HPI-software monitoring could potentially reduce the duration and severity of intraoperative hypotension in patients undergoing noncardiac surgery. Despite these findings, the current literature does not conclusively establish whether the prevention of hypotension leads to a decrease in cardiac stress, as evidenced by the dynamics of cardiac biomarkers.

This study aimed to determine the dynamics of troponin and NTproBNP concentration changes during the first two postoperative days in patients undergoing high-risk cancer abdominal surgery with advanced hemodynamic monitoring featuring HPI. The hypothesis posited that the individualized hemodynamic management and active, targeted prevention of hypotension could reduce cardiovascular complications (PMI, AKI, stroke). A secondary objective was to explore the relationship between biomarker dynamics and hypotension or hemodynamic instability, indicated by an HPI value exceeding 85%.

## 2. Material and Methods

### 2.1. Study Design

This prospective study was conducted at the Mazovia Regional Hospital, a tertiary oncologic center, from March 2023 to October 2023, involving a cohort of 50 consecutive patients scheduled for major cancer gastrointestinal surgery who met the following conditions:

Age: Participants had to be older than 18 years of age.

ASA classification: Patients classified under the American Society of Anesthesiologists Physical Status Classification System (ASA) with a status of 3 or 4 were eligible for inclusion.

Perioperative risk profile: Procedures of intermediate (≥1 to 5%) or high (>5%) surgical risk of postoperative cardiovascular complications [9] qualified.

Surgical procedure: Eligible participants were those scheduled for oncological gastrointestinal surgery.

Perioperative hemodynamic monitoring: Perioperative hemodynamic monitoring was conducted using the Acumen^TM^ IQ sensor and the HemoSphere monitoring platform (both Edwards Lifesciences, Irvine, CA, USA)

Informed consent: Patients who provided informed consent for participation in this study qualified for inclusion.

Approval for this study was procured from the Bioethics Committee at the Regional Chamber of Physicians and Dentists in Warsaw under the reference number KB 1421/22.

### 2.2. Clinical Definitions

Advanced hemodynamic parameters were obtained using the HemoSphere monitoring platform, equipped with the Acumen^TM^ IQ sensor, through minimally invasive access via a radial artery line. In addition to directly measured values obtained via an arterial line, such as arterial pressure and heart rate, the Edwards HemoSphere device utilizes specific advanced algorithms to derive a series of hemodynamic parameters. These parameters are based on the characteristics of the arterial waveform, including its shape, amplitude, and temporal changes. The derived parameters include the following [25]:

The cardiac index (CI) quantifies the cardiac output relative to body surface area, measured in liters per minute per square meter.

Stroke volume (SV) denotes the volume of blood ejected by the left ventricle during a single contraction.

The Systemic Vascular Resistance Index (SVRI) provides a normalized measure of the resistance of the systemic vasculature, adjusted for body surface area.

Stroke volume variation (SVV) indicates the fluctuation in stroke volume with each cardiac cycle.

The Rate of Change of Left Ventricular Pressure in time (dP/dt) measures the rate of pressure change in the left ventricle during systole and is a key indicator of cardiac contractility.

Dynamic Vascular Elastance (Eadyn) is a parameter indicating the responsiveness of the arterial system to changes in stroke volume. It helps in understanding the interaction between the heart and the vascular system, especially in guiding fluid therapy and vasopressor use.

The HPI (Hemodynamic Performance Index) is an advanced algorithm-based parameter that combines various hemodynamic measurements to predict the likelihood of a patient developing hypotension.

An episode of hypotension was defined as a decrease in MAP below 65 mmHg for more than 1 min. The HemoSphere monitoring platform acquired various quality indices of hypotension in real-time, which included the following:The number of hypotension episodes.Total duration of hypotension episodes (unit: min).Average duration of each hypotension episode (unit: min).The proportion of time spent in hypotension (MAP < 65 mmHg) relative to the total monitoring duration (unit: %).Area Under the Threshold (AUT) (unit: mmHg × min).Time-weighted average (TWA) of MAP < 65 mmHg, calculated as the area under the curve of MAP below 65 mmHg divided by the total monitoring time (unit: mmHg).Number of hypotension episodes with MAP < 50 mmHg.Number of episodes with an increase in the HPI value above 85.

Perioperative myocardial injury and infarction (PMI) was defined in accordance with the 2022 ESC guidelines for NCS as acute cardiomyocyte injury (postoperative high-sensitivity cardiac Troponin T/I release) with or without accompanying symptoms and with or without ECG or imaging evidence of acute myocardial ischemia [9]. In biochemical studies, this was characterized by an absolute increase in high-sensitivity cardiac Troponin T/I (e.g., the 99th percentile upper reference limit) above the preoperative high-sensitivity cardiac Troponin T/I concentration [9].

Acute kidney injury (AKI) was defined in accordance with the Kidney Disease: Improving Global Outcomes Clinical Practice Guideline for Acute Kidney Injury (KDIGO) criteria. AKI was diagnosed when the serum creatinine level increased by at least 0.3 mg/dl within 48 h post-operation relative to the baseline value or when there was an increase in the serum creatinine level of more than 50% from the baseline value during the first 7 days post-operation [26].

### 2.3. Perioperative Management

Patients were prepared for anesthesia and surgery based on their health status, comorbidities, and the planned operation. Advanced hemodynamic monitoring using the Acumen IQ sensor in conjunction with the HemoSphere platform and HPI technology was initiated prior to the induction of general anesthesia. This enabled the acquisition of the HPI parameter and other advanced cardiovascular dynamics parameters, including CI, SV, SVRI, SVV, dP/dt, and Eadyn (Figure 1). Anesthesia induction and maintenance were conducted according to the clinical routine of the attending anesthesiologist. After typical preoxygenation, the intravenous induction of general anesthesia was performed. Patients received fentanyl (2 mcg/kg), propofol (1.5–2.5 mg/kg, individually adjusted based on age and coexisting conditions), and rocuronium (0.6 mg/kg) for muscle relaxation. Atropine was administered during induction if indicated. Anesthesia maintenance was achieved using desflurane (MAC: 1.0–1.1) and fractional doses of rocuronium and fentanyl. Radial artery cannulation was performed after administering local anesthesia with 1 mL of 2% lignocaine (subcutaneous administration). Hemodynamic monitoring continued until the patient left the operating room. The anesthesiologists, experienced in administering general anesthesia and operating the HemoSphere platform, had received official training in the use of HPI software with the Acumen IQ sensor and the HemoSphere platform. Patient management was adapted according to the clinical routine and HPI training of the attending anesthesiologist. The algorithms trained on the Edwards Lifesciences platform, utilizing the HPI, recommend interventions when the HPI value exceeds 85. This is to prevent a drop in MAP based on current hemodynamic parameters [19,25]. Specifically, if a patient demonstrates fluid responsiveness (defined as SVV ≥ 13%) and has an Eadyn above 1.1, administering a fluid bolus of a 250 mL crystalloid is advised to increase MAP. Conversely, if SVV is 13% or higher, but Eadyn is below 1, the use of vasopressors is recommended to maintain hemodynamic stability. In cases where dP/dt is below 400, despite a normal SVV, inotropes should be administered. Furthermore, if both SVV and dP/dt are within normal ranges, but the HPI exceeds 85, an increase in the Systemic Vascular Resistance Index (SVRI) is suggested (Figure 1). Monitoring and data recording using the HPI software began before the induction of general anesthesia and concluded upon the patient’s departure from the operating room. Serum Troponin T, NTproBNP levels, and electrocardiograms were used preoperatively, then again at 24 and 48 h post-operation.

### 2.4. Data Collection

The analysis was based on data obtained from the patient’s medical records, including the anesthesia qualification form, surgical records, anesthesia records, postoperative monitoring records, and physician’s orders. Hypotension data, including the number of hypotension episodes, total duration of hypotension episodes, average duration of a hypotension episode, AUT, TWA for MAP < 65 mmHg, number of hypotension episodes with MAP < 50 mmHg, and number of episodes with an increase in the HPI value above 85, were obtained using the dedicated Acumen Analytics v. 2.0.0. software (Edwards Lifesciences, Irvine, CA, USA).

### 2.5. Statistical Analysis

The delta between parameter values was calculated as the difference between the later measurement and the earlier one (measurement at 24 h postoperatively minus the baseline measurement; measurement at 48 h postoperatively minus the baseline; measurement at 48 h postoperatively minus the measurement at 24 h postoperatively). The percentage change was calculated as the ratio of the later measurement to the earlier one, multiplied by 100%. Higher values indicate a benefit for the later parameter (its value relative to the earlier measurement was higher), whereas negative values indicate a decrease in the parameter value compared to the earlier measurement.

For quantitative variables, basic descriptive statistics were calculated with the Shapiro–Wilk test to assess the normality of the distribution. In order to compare the groups in terms of quantitative variables, the Mann–Whitney U test was utilized. The Pearson chi-squared test (for independence) or Fisher’s exact test (when the expected number was less than 5) was applied for categorical data. To determine the relationships between quantitative/ordinal variables, the Spearman’s rho correlation coefficient was used. Because the distributions of NT-proBNP and troponin were skewed, a non-parametric Friedman test was used to test for changes between preoperative, 24 h postoperative, and 48 h postoperative NT-proBNP and troponin values, and the Wilcoxon-signed-rank test for a within-patient change in NT-proBNP and troponin. A significance level of *p* < 0.05 was adopted for all analyses. Values of p between 0.05 and 0.10 were considered to indicate a trend, but not statistically significant. The statistical analyses were performed using IBM SPSS Statistics 28.0 software.

## 3. Results

### 3.1. Patients’ Characteristics

A total of 50 consecutive eligible patients were enrolled in this study. Four patients were excluded from the final analysis due to the lack of all required laboratory tests—consequently, the analysis was carried out for 46 patients. The majority of the study group were men (69.6%). The average age of patients was nearly 70 years, with the oldest being 86. The most common chronic diseases were hypertension (69.6%) and chronic heart failure (45.7%). Regarding the ASA scale, 65.2% of patients were classified as ASA3, and 34.8% were in the ASA4 group. The POSSUM scale scores for the patients were high, with an average risk of postoperative complications being 75.66% and a risk of postoperative death being 34.58%. More than half of the patients (54.3%) underwent preoperative chemotherapy treatment, and gastrectomy was the most common surgery performed (43.5%). Half of the procedures (50%) lasted over 4 h, and almost all patients (97.8%) required perioperative use of noradrenaline. The average initial level of NTproBNP was 641 pg/mL, with values ranging from 12.3 to 13,350.4 pg/mL. The average initial troponin level was 0.0158 μg/L, which exceeded the laboratory norm’s 99th percentile upper reference limit (n < 0.014 μg/L). The full characteristics of the study group are presented in Table 1.

Most patients in this study (N = 38, 82.6%) experienced at least one episode of hypotension. Among the eight patients who did not experience hypotension, there was a median of eight HPI alarms per surgery, suggesting effective preemptive interventions facilitated by this technology. The remaining 38 patients had an average of two hypotension episodes each, with a median of 11 HPI alarms. The median TWA of MAP < 65 mm Hg was 0.085 (IQR: 0183) mm Hg. The median number of ≥1 min episodes of a MAP < 65 mm Hg was 2 (IQR: 2.25). Patients spent a median of 1.95 (IQR: 1.87) min below MAP < 65 mmHg. The median number of ≥1 min episodes of a MAP < 50 mm was zero (Table 2). Nearly all patients (97.8%) required vasopressor support with noradrenaline, and more than one-third (37%) required a combined regimen of noradrenaline and dobutamine (Table 1).

In this cohort, PMI was observed in 5 patients (10.9%), and AKI occurred in 11 patients (23.9%) within two days postoperatively. Additionally, one patient (2.2%) exhibited postoperative symptoms of acute respiratory failure, and four patients (8.7%) presented with HF symptoms necessitating further pharmacological management. Notably, one patient (2.2%) required hospital readmission within 30 days following surgery, and there was a single instance of mortality (2.2%) within the same postoperative period. Table 3 delineates the relationship between various hypotension-related parameters and patient groups, specifically those with postoperative PMI or AKI and those with preoperatively diagnosed chronic HF. A pronounced incidence of at least one hypotension episode was noted across all cohorts, with the HF group exhibiting the greatest frequency (95% vs. 72% in patients without HF, *p* = 0.055). A significant finding was that patients with AKI were more likely to experience prolonged episodes (≥1 min) of MAP < 50 mmHg (11 patients with AKI had seven such episodes, in contrast to eight episodes among the 35 patients without AKI; *p* = 0.027). No other significant differences were identified despite the observed variability among the different patient groups. This lack of a significant difference could be attributed to the small sample sizes, potentially restricting the statistical power necessary to discern more subtle distinctions in hypotension parameters between the groups. These findings imply that either the true differences in hypotension parameters among these groups are not pronounced or that this study may lack sufficient power to detect them.

### 3.2. Perioperative Dynamics of NTproBNP

NTproBNP plasma concentration for the entire patient cohort was 183.9 [interquartile range (IQR): 428.1] pg/mL at baseline, increasing to 285.8 [IQR: 679.8] pg/mL at 24 h, and 276.7 [IQR: 609.4] pg/mL at 48 h post-surgery (*p* < 0.001). A statistically significant elevation in NTproBNP levels was noted when comparing preoperative values to those measured 24 h after surgery (*p* < 0.001), as well as at 48 h postoperatively (*p* = 0.002). However, the change in NTproBNP levels between 24 and 48 h post-surgery was not statistically significant (*p* = 0.154). Postoperatively, an increase in NT-proBNP was observed in 80.4% of patients. The absolute and relative increases in NTproBNP post-surgery were reported as 91.2 [IQR: 177.4] pg/mL (median, IQR) and 177.6% at 24 h, and 64.6 [IQR: 241] pg/mL and 166% at 48 h, respectively.

When analyzing data with respect to preoperative chronic heart failure, it was demonstrated that patients with diagnosed heart failure exhibited significantly higher baseline levels of NTproBNP (435 [IQR: 711.15] pg/mL, *p* < 0.001), as well as at 24 h (482.3 [IQR: 1067.6] pg/mL, *p* = 0.002) and 48 h (346 [IQR: 1180.2] pg/mL, *p* = 0.006) postoperatively, compared to patients without diagnosed heart failure (Table 4). Among the heart failure cohort (n = 21), the postoperative change in NTproBNP was significant (*p* = 0.023), with significance only between the preoperative and 24 h postoperative measurements (*p* = 0.023). In the non-heart-failure group (n = 25), the postoperative change in NTproBNP was also significant (*p* < 0.001), both between preoperative values and 24 h post-surgery (*p* = 0.001) and between preoperative values and 48 h post-surgery (*p* = 0.007), with no statistical significance in the dynamic changes between the first and second postoperative days. In patients without recognized preoperative chronic heart failure, the percentage difference in NTproBNP levels between baseline and 24 h post-surgery was greater compared to patients with a heart failure diagnosis (225.17% vs. 146.28%, *p* = 0.032). Table 4 presents the NTproBNP level changes for the entire study group, marked preoperatively, at 24 and 48 h post-surgery. The changes in NTproBNP levels over a 48 h period post-surgery were also analyzed in relation to the co-occurrence of chronic diseases, with no significant differences observed (see Appendix A). Additionally, no correlation was found between the dynamics of NT-proBNP levels and scores on the POSSUM scale (Appendix A).

### 3.3. Perioperative Dynamics of Troponin

Median troponin concentrations for the entire patient cohort were observed at 0.012 [IQR: 0.011] μg/L at baseline, increasing to 0.017 [IQR: 0.01075] μg/L at 24 h, and 0.0165 [IQR: 0.013] μg/L at 48 h post-surgery (*p* < 0.003). A statistically significant elevation in troponin concentration was observed when comparing preoperative values to those measured 24 h after surgery (*p* = 0.001), as well as at 48 h postoperatively (*p* < 0.001). The change in troponin concentration between 24 and 48 h post-surgery was not statistically significant (*p* = 0.767) (see Figure 2). In total, 39.1% (n = 18) of patients had an elevated baseline level of troponin. Postoperatively, an increase in troponin levels was observed in 67% of patients, and 11% (n = 5) met the criteria for perioperative myocardial injury. However, due to the absence of electrocardiogram changes, the criteria for myocardial infarction were not fulfilled. Patients with higher preoperative troponin levels experienced greater hemodynamic instability, as indicated by a higher number of HPI alarms (14 vs. 9), although this difference was not statistically significant (*p* = 0.313). Nevertheless, the total number of hypotension episodes and the mean duration of these episodes were similar between both groups (Appendix A).

The analysis assessed the relationship between NTproBNP change dynamics and PMI occurrence (Appendix A). The medians of NT-proBNP levels were higher in the PMI group, with the largest increase observed on the first postoperative day; however, this difference was not statistically significant (273 [IQR: 421.85] vs. 1381 [IQR: 3136.75] pg/mL in PMI, *p* = 0.315). Additionally, this study found no significant difference in the frequency or duration of hypotension episodes between patients with and without PMI (Table 3).

### 3.4. Perioperative Dynamics of NtproBNP and Acute Kidney Injury

In the study group, the frequency of acute kidney injury (AKI) was 23.9% (n = 11), with 10 patients classified in KDIGO class 1, and only 1 patient in class 2. None of the patients required renal replacement therapy. Patients who developed AKI had significantly higher baseline NT-proBNP levels (435 [IQR: 926.6] pg/mL vs. 111.4 [IQR: 294] pg/mL, *p* = 0.036). In these patients, a reduction in the level of this biomarker was observed over the two subsequent postoperative days; however, the change between preoperative and postoperative values was not statistically significant (*p* = 0.790), with a significant decrease noted between the first and second postoperative day (*p* = 0.006). This trend was completely different from the changes in NTproBNP levels in patients who did not develop AKI. These patients experienced a significant postoperative increase in NTproBNP concentration (*p* < 0.001), specifically between preoperative values and 24 h post-surgery (*p* < 0.001), and 48 h post-surgery (*p* < 0.001), but without a significant change between the first and second postoperative days (*p* = 1.0) (Figure 3 and Appendix A).

### 3.5. Perioperative Biomarkers’ Dynamic in Laparoscopic vs. Open Surgery

In total, 36.9% of the patients (n = 17) underwent minimally invasive robotic or laparoscopic surgeries in the steep Trendelenburg position (28–32 degrees). No differences were observed in the occurrence of hypotension, frequency of PMI, and AKI between these groups (Appendix A). However, a statistically significant difference was noted in the dynamics of NT-proBNP—the absolute increases in the concentration of this biomarker in the blood of patients were substantially higher in those undergoing laparoscopic surgeries (absolute and relative change: 192.2 [IQR: 592.4] pg/mL and 177.35% vs. 62.25 [IQR: 103.83] pg/mL and 168.53%, *p* = 0.031). No differences in the dynamics of troponin levels were noted.

## 4. Discussion

Abdominal oncologic surgery presents unique challenges and hazards related to an increased perioperative risk on the patient and procedural sides. Patients with cancer commonly preoperatively present with weakened health and multiple comorbidities, including cardiovascular risk factors, malnutrition, and compromised immune systems, making them more susceptible to anesthesia-related adverse events [9]. In a 2018 report from the USA National Inpatient Sample database, nearly half (44.5% of 10581621 hospitalizations for major NCS) of adults aged ≥45 years undergoing major noncardiac surgery presented with at least two major cardiovascular risk factors. Moreover, the burden of cardiovascular risk factors and their prevalence increased over time [27]. In particular, cancer surgeries can be prolonged and procedurally complicated, carrying an increased risk of the procedure itself, such as significant blood loss, deep vein thrombosis, pressure sores, and respiratory or cardiovascular instability [28].

The minimum risk associated with major adverse cardiac events in oncologic abdominal surgeries, such as colorectal surgery with bowel resection, is estimated to be high (>5%) and rises further with the increased complexity of the procedure, as in duodenal–pancreatic surgery, liver resection, or esophagectomy [8,9].

The 2014 ACC/AHA Guideline on Perioperative Cardiovascular Evaluation and Management for noncardiac surgery categorizes elevated-risk procedures when the combined surgical and patient risk of major adverse cardiac events is ≥1 percent, therefore encompassing most abdominal cancer surgeries as elevated-risk procedures [8]. Furthermore, previous cancer treatments, such as chemotherapy or radiation, can affect both the surgical and anesthesia processes, adding to the complexity of managing fluid balance and blood loss during these extensive surgeries [29,30].

This complexity, coupled with the way anesthesia can influence hemodynamic stability, underscores the need for highly skilled and tailored anesthesia management to mitigate the risks of hypotension or cardiac events in this patient population. Research primarily based on retrospective and observational studies, including meta-analyses, indicates that a MAP below 60 mmHg is associated with increased organ injury and mortality. These findings were foundational for the Perioperative Quality Initiative consensus statement on intraoperative blood pressure, published in 2019 [31], which concluded that an intraoperative MAP below 60–70 mm Hg is associated with myocardial injury, acute kidney injury, and death, noting that the extent of injury correlates with the severity and duration of hypotension. Although the optimal intraoperative blood pressure target to minimize perioperative complications remains uncertain [32], personalizing intraoperative MAP targets could significantly improve perianesthesia outcomes. However, this strategy requires further investigation to establish its effectiveness [33]. Utilizing HPI in hemodynamic monitoring software, which employs predictive analytics, has been shown to not only forecast and potentially prevent episodes of hemodynamic destabilization but also to facilitate achieving targeted blood pressures intraoperatively. Current research indicates that HPI can effectively reduce the occurrence, duration, and severity of IOH in noncardiac surgeries, particularly when standard intraoperative protocols are adhered to properly [20,21,22,23,24,34,35,36,37].

In our analysis, hypotension—defined as at least one episode of MAP < 65 mmHg lasting ≥1 min—was common, occurring in over four-fifths of the patients (82.6%). However, comparisons with other studies are challenging due to variations in how IOH is reported and measured, such as invasive versus manual monitoring and real-time versus interval-based assessments [38]. For instance, Sessler and colleagues [39] reported a 69% incidence of IOH in general noncardiac surgery among 25,702 patients, while Gregory et al. [35] observed it in 19.3% of 70,938 surgical cases, where blood pressure was measured at intervals. In contrast, data from the recent prospective observational EU Hyproject registry [24] showed that 59.4% of patients in ASA classes 1–4 experienced a MAP < 65 mm Hg for at least 1 min when assessed in real-time using the Acumen^TM^ IQ sensor and the HemoSphere monitoring platform by Edwards Lifesciences. Compared to these studies, our cohort, predominantly of higher-risk patients classified as ASA class 3 (65.2%) or 4 (34.8%), exhibited a higher incidence of IOH. Despite the high categorical rates of experiencing MAP < 65 mm Hg for at least 1 min, paralleling EU Hyproject registry [24] results, the application of HPI software version 02.03.000.103 revealed that the median time-weighted average MAP < 65 mm Hg in our cohort was notably low, at 0.085 (IQR: 0.03–0.19) mmHg with a median of 3 min spent below a MAP of 65 mmHg.

Notably, these episodes were more prevalent in patients with heart failure (95.2%) compared to those without (72.0%), although this difference was not statistically significant (*p* = 0.055). There were no significant differences observed in TWA MAP < 65 mmHg (0.1 [IQR: 0.03–0.19] mm Hg vs. 0.07 [IQR: 0.03–0.19] mm Hg, respectively) or in the median time spent below 65 mm Hg (3 min for both groups). However, patients with heart failure triggered more HPI alarms, leading to more frequent interventions by anesthesiologists (11 vs. 9).

These findings suggest that incorporating HPI into perioperative care facilitates proactive management to achieve stable hemodynamics. It assists in anticipating and mitigating episodes of low blood flow, thereby reducing cardiac workload and potentially decreasing the release of cardiac stress markers such as NTproBNP.

NTproBNP’s utility extends beyond a heart failure diagnosis to various clinical contexts. While elevated BNP or NTproBNP might also indicate undiagnosed heart failure, they also rise in pulmonary diseases (e.g., pulmonary embolism, obstructive sleep apnea, chronic obstructive lung disease), neurological disorders (e.g., stroke), renal insufficiency, and other critical illnesses. However, these noncardiac conditions can induce BNP or NT-pro-BNP elevation through underlying cardiac issues [9,11,12,13,14,15,16].

In particular, preoperative NT-pro-BNP elevation is strongly linked to increased postoperative cardiovascular event risk [9,15,40]. However, interpreting postoperative NT-proBNP dynamics is more complex. An 18-study meta-analysis (n = 2179) examining pre- and postoperative Natriuretic Peptide levels found that including postoperative levels in a risk model enhanced 30-day and ≥180-day risk classification. Elevated postoperative levels, particularly above 718 (95% CI: 656–994) pg/mL, were key independent outcome predictors at both intervals [16]. In our study, the median postoperative NT-pro-BNP values were 201 (IQR: 268) pg/mL for patients without diagnosed heart failure and 482 (IQR: 1067) pg/mL for those with heart failure. These levels in both groups were below the risk threshold identified by Redseth et al. Furthermore, the average of 10 HPI alarms triggering anesthesiologist intervention per surgery in our cohort suggests the effective mitigation of detrimental volume status changes and hypotension episodes. Consequently, this individualized hemodynamic management using HPI may have contributed to reduced cardiac stress expressed as lower NT-pro-BNP elevations.

Conversely, another study examining NT-pro-BNP dynamics in young, healthy patients undergoing noncardiac surgery reported lower postoperative NTproBNP levels (median of 42 [11–86] pg/mL) and found no correlation with adverse clinical events [41]. This pattern is somewhat mirrored in our study, where patients without heart failure exhibited lower postoperative NTproBNP levels (201 pg/mL) compared to those with heart failure (482 pg/mL). However, in both groups, the postoperative shift in NTproBNP was not statistically significant when compared to preoperative values (70.9 vs. 104 pg/mL, respectively).

This suggests a link to perioperative volume status changes, indicating that subtle subsequent NTproBNP elevations may not always signify critical cardiac stress or predict adverse outcomes, particularly in healthy populations. Conversely, in patient groups with existing circulatory clinical or subclinical dysfunctions or other significant comorbidities, these transient changes in volume status could induce more substantial cardiac stress, as reflected in more pronounced NTproBNP kinetics, potentially leading to subsequent adverse events, if hemodynamic instability is not managed early. Therefore, the role of postoperative NTproBNP as a marker of impaired perioperative hemodynamics merits consideration.

This conclusion is supported by the finding that when using HPI in our study of high-risk patients, the incidence of PMI of 11% was comparable to other studies [9]. However, no association was found between PMI and the occurrence of hypotension in all cohorts, with similar levels of hypotension parameters observed regardless of PMI status. This raises the question of whether intraoperative hypotension in patients with heart failure indicates impaired circulatory autoregulation or disease severity rather than directly causing postoperative complications [32,42]. Furthermore, no neurological complications were associated with hypotension. Conversely, patients meeting the AKI criterion had more episodes of MAP < 50 mmHg than those who did not, although other hypotension parameters were consistent across both groups. This observation aligns with the concept of organ blood flow autoregulation, which maintains constant blood flow within certain MAP ranges, considered to be 50–160 mmHg for the brain [43], 70–130 mmHg for the kidneys [44], and 60–140 mmHg [45] for the myocardium. This suggests that different pressure threshold goals could be taken into consideration for the kidneys compared to the brain and heart.

## 5. Conclusions and Limitations

Our study found minimal intraoperative hypotension, as evidenced by brief and mild episodes of mean arterial pressure falling below 65 mmHg. Advanced hemodynamic monitoring with HPI was effective in reducing significant intraoperative hypotension episodes, both in patients with and without preoperative heart failure or elevated troponin levels. Notably, high preoperative NT-pro-BNP levels were associated with an increased risk of acute kidney injury, and laparoscopic surgery patients showed significant NT-pro-BNP level increases postoperatively. These results indicate that HPI technology can achieve hemodynamic stability in high-risk patients (ASA classes 3 and 4), reducing hypotension episodes.

This study has several limitations that should be considered when interpreting the findings. Firstly, the small sample size potentially limits the statistical power of the analysis, making it challenging to draw definitive conclusions or generalize the results to a broader population. Secondly, as a single-center study, the results might be influenced by specific practices or patient demographics unique to this center, which may not represent other settings. Thirdly, the absence of a control group in the study design limits the ability to assess the effectiveness of interventions compared to standard care or other alternative treatments. These limitations highlight the need for larger, multi-center studies with control groups to validate and expand upon our findings.

## Figures and Tables

**Figure 1 jpm-14-00211-f001:**
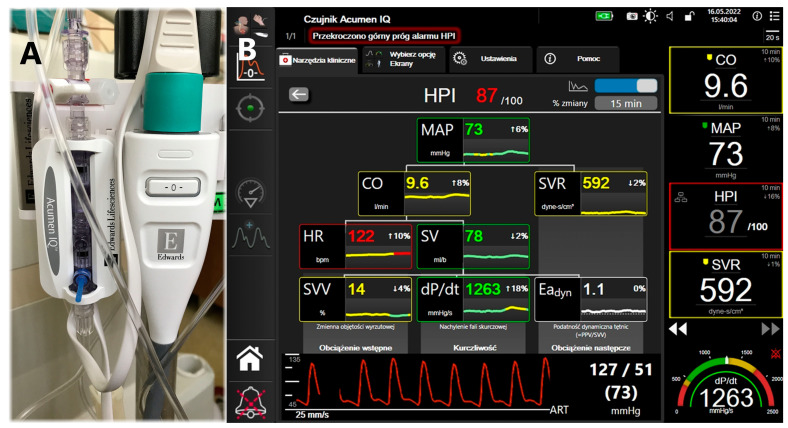
(**A**) shows an Acumen IQ minimally invasive sensor connected through radial artery line. (**B**) is an example secondary screen of derived hemodynamic parameters with HPI technology. In this example, with an HPI above 85, SVV at 14%, and Eadyn at 1.1, administering fluid therapy is the suggested response to prevent hypotension.

**Figure 2 jpm-14-00211-f002:**
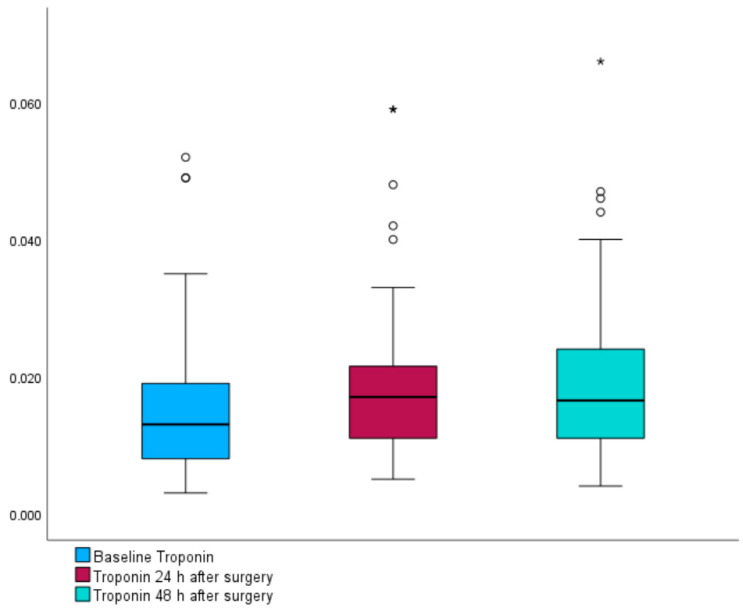
Preoperative and postoperative troponin levels. Boxplots illustrate the median and interquartile range (IQR) of troponin concentrations at baseline and 24 and 48 h after surgery. The whiskers extend to the furthest points within 1.5 times the IQR from the upper and lower quartiles, as per Tukey’s method. The asterisk (*) denotes outliers.

**Figure 3 jpm-14-00211-f003:**
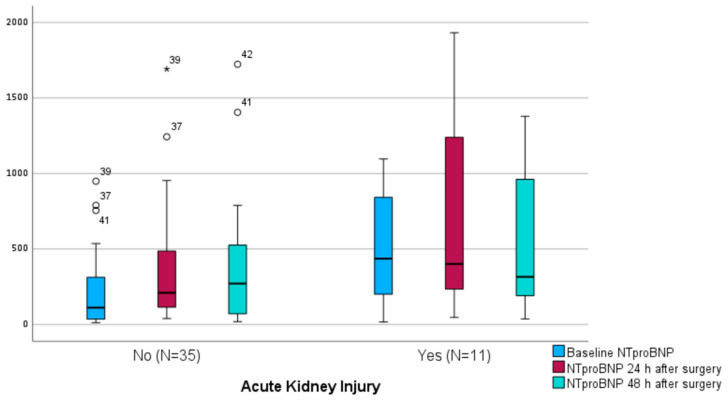
Preoperative and postoperative NtproBNP levels stratified by postoperative acute kidney injury. Boxplots illustrate the median and interquartile range (IQR) of NTproBNP levels at baseline and 24 and 48 h after surgery. The whiskers extend to the furthest points within 1.5 times the IQR from the upper and lower quartiles, as per Tukey’s method. The asterisk (*) denotes outliers.

**Table 1 jpm-14-00211-t001:** Patients’ Characteristics.

Variable	Result
Gender	
Female, n (%)	14 (30.4%)
Male, n (%)	32 (69.6%)
Age, Years, Mean (SD), (Min–Max)	69.83 (11.61), (36–86)
Chronic Diseases	
Hypertension, n (%)	32 (69.6%)
Coronary Heart Disease, n (%)	16 (34.8%)
Chronic Obstructive Pulmonary Disease, n (%)	1 (2.2%)
Asthma, n (%)	3 (6.5%)
Diabetes, n (%)	15 (32.6%)
Atherosclerosis, n (%)	19 (41.3%)
Obesity, n (%)	15 (32.6%)
Chronic Heart Failure, n (%)	21 (45.7%)
Chronic Kidney Disease, n (%)	10 (21.7%)
ASA Score	
3, n (%)	30 (65.2%)
4, n (%)	16 (34.8%)
Preoperative Oncological Treatment	
Chemotherapy, n (%)	25 (54.3%)
Radiotherapy, n (%)	6 (13.0%)
Surgery Type	
Right Hemicolectomy, n (%)	9 (19.6%)
Left Hemicolectomy, n (%)	3 (6.5%)
Anterior Resection of Rectum, n (%)	7 (15.2%)
Gastrectomy, n (%)	20 (43.5%)
Whipple Operation, n (%)	6 (13.0%)
Cardia Resection, n (%)	1 (2.2%)
Surgery Duration, Minutes, Mean (SD), (Min−Max)	241.93 (97.10), (55–485)
Surgery Duration Interval	
<2 h, n (%)	4 (8.7%)
2–4 h, n (%)	19 (41.3%)
>4 h, n (%)	23 (50.0%)
Type	
Open, n (%)	29 (63.0%)
Laparoscopic, n (%)	17 (37.0%)
POSSUM Score	
Physiological Risk, Mean (SD), (Min−Max)	26.76 (8.55), (14–54)
Surgical Risk, Mean (SD), (Min−Max)	17.17 (5.55), (8–33)
Morbidity, %, Mean (SD), (Min−Max)	75.66 (21.79), (23.33–99.69)
Mortality, %, Mean (SD), (Min−Max)	34.58 (23.78), (4.07–92.69)
Total Hospitalization Time, Days, Mean (SD), (Min−Max)	18.74 (11.48), (6–58)
Postoperative Hospitalization Time, Days, Mean (SD), (Min−Max)	12.96 (9.02), (4–51)
Intraoperative Use of Inotropes or Vasopressors	
Noradrenaline, n (%)	45 (97.8%)
Dobutamine, n (%)	17 (37.0%)
Combined use of	
Noradrenaline and Dobutamine, n (%)	17 (37.0%)
Baseline NTproBNP Level, pg/mL, Mean (SD), (Min−Max)	641.33 (1994.89), (12.3–13,350.4)
Baseline Troponin Level, μg/L, Mean (SD), (Min−Max)	0.0158 (0.01), (0.003–0.052)
Baseline Creatinine Level, mg/dl, Mean (SD), (Min−Max)	1.05 (0.57), (0.61–4.49)

Abbreviations: SD—standard deviation, Min–Max—minimum and maximum values for a given parameter, ASA—American Society of Anesthesiologists scale, POSSUM—The Physiological and Operating Severity Scale for Mortality and Morbidity.

**Table 2 jpm-14-00211-t002:** Comparison of Hypotension Parameters in Patients with and without Intraoperative Hypotension Episodes.

	All Patients (n = 46)		Patients without Hypotension Episodes (n = 8)		Patients with Minimum of One Hypotension Episode (n = 38)		
Variable	Median	IQR	Median	IQR	Median	IQR	*p*-Value
Total duration of monitoring (min)	262.5	147.8	225	164.5	271.5	123.8	0.434
Number of episodes with HPI > 85	9.5	11	8	3.25	11	12	0.12
The number of hypotension episodes	2	2.25	0	0	2	3.25	<0.001
Total duration of hypotension episodes, minutes	3	6.5	0	0	5	9.03	<0.001
The proportion of time spent in hypotension, %	1.31	3.18	0	0	1.67	3.66	<0.001
Average duration of each hypotension episode, min	1.95	1.87	0	0	2	1.44	<0.001
Mean MAP if < 65 mmHg	60.07	3.09	0	0	60.07	3.09	<0.001
AUT < 65 mmHg	23.67	53.92	0	0	30.17	53.25	<0.001
TWA MAP < 65 mmHg	0.0850	0.183	0	0	0.12	0.25	<0.001
Number of episodes with MAP < 50 mmHg	0	0	0	0	0	0	0.194

Abbreviations: MAP—Mean Arterial Pressure; HPI—Hypotension Prediction Index (HPI) Value; AUT—Area Under the Threshold; TWA—Time-Weighted Average; IQR—Interquartile Range.

**Table 3 jpm-14-00211-t003:** Impact of Preoperative chronic heart failure and postoperative complications on intraoperative hypotension Parameters.

	Preoperative					Postoperative									
	HF					AKI					PMI				
	No (n = 25)		Yes (n = 21)			No (n = 35)		Yes (n = 11)			No (n = 41)		Yes (n = 5)		
Variable	Mdn	IQR	Mdn	IQR	*p*-Value	Mdn	IQR	Mdn	IQR	*p*-Value	Mdn	IQR	Mdn	IQR	*p*-Value
Min 1 episode of hypotension, n (%)	18 (72%)		20 (95%)		0.055	28 (80%)		10 (91%)		0.658	35 (85%)		3 (60%)		0.203
Total duration of monitoring (min)	279	120.17	250	177.5	0.225	279	137	230	134	0.067	259	126.5	357	169.5	0.298
Number of episodes with HPI > 85	9	9	11	12	0.347	10	9	6	14	0.409	10	11.5	8	10	0.873
The number of hypotension episodes	2	4	2	2	0.374	2	2	1	6	0.844	2	2.5	2	7	0.914
Total duration of hypotension episodes, min	3	9.85	3	4.5	0.564	3	6	2	12	0.806	3	6.5	3	35.35	0.696
The proportion of time spent in hypotension, %	1.03	3.11	1.4	3.5	0.32	1.21	3.2	1.4	5.53	0.353	1.35	3.12	0.84	10.01	0.764
Average duration of each hypotension episode, min	2	2.92	1.89	1.92	0.642	1.89	1.83	2	3.35	0.332	2	1.83	1.5	3.9	0.376
Mean MAP if < 65 mmHg	60.07	4.07	60.32	2.88	0.483	60.19	2.28	58.62	6.24	0.233	60.14	2.95	59.04	−56.03	0.607
AUT < 65 mmHg	22.67	61.84	24.67	68.34	0.674	21.67	44.67	28	102.34	0.366	24.67	55.17	11.33	323.17	0.559
TWA MAP < 65 mmHg	0.07	0.2	0.1	0.29	0.426	0.07	0.17	0.11	0.75	0.25	0.09	0.17	0.03	0.91	0.645
Number of episodes with MAP < 50 mmHg	0	0	0	0	0.916	0	0	0	1	0.027	0	0	0	1.5	0.672

Abbreviations: MAP—Mean Arterial Pressure; HPI—Hypotension Prediction Index (HPI) Value; AUT—Area Under the Threshold; TWA—Time-Weighted Average; Mdn—Median; IQR—Interquartile Range; PMI—Perioperative Myocardial Injury; AKI—Acute Kidney Injury; HF—Preoperative Heart Failure.

**Table 4 jpm-14-00211-t004:** Perioperative NTproBNP Level Dynamics in Patients with and without Chronic Heart Failure.

	All Patients				Heart Failure		No Heart Failure				
					N = 21		N = 25				
	Mdn	IQR	Min.	Max.	Mdn	IQR	Mdn	IQR	Z	*p*-Value	r
NT-pro-BNP–baseline	183.9	428.1	12.3	13,350.4	435.9	711.15	87	232.2	−3.34	<0.001	0.49
NT-pro-BNP–24 h	285.8	679.8	39.0	10,558.5	482.3	1067.6	201.4	268.1	−3.08	0.002	0.45
NT-pro-BNP–48 h	276.7	609.4	19.2	17,067.9	346	1180.2	157.6	377.05	−2.72	0.006	0.4
Δ NT-pro-BNP–1-0	91.2	177.4	−2791.9	1959.7	104.6	292.7	70.9	150.4	−0.69	0.487	0.1
Δ NT-pro-BNP–2-0	64.6	241.0	−4743.6	14,793.8	90.1	290.35	37	192.6	−0.45	0.651	0.07
Δ NT-pro-BNP–2-1	−31.3	160.1	−1951.7	13,034.8	−60.6	325.55	−19.8	110.3	−0.69	0.487	0.1
%Δ NT-pro-BNP–1-0	177.6	171.8	64.1	658.8	146.28	117	225.17	267.82	−2.15	0.032	0.32
%Δ NT-pro-BNP–2-0	166.0	190.7	28.5	793.2	153.1	145.86	226.99	269.58	−1.16	0.247	0.17
%Δ NT-pro-BNP–2-1	82.7	52.6	17.5	658.5	81.52	80.74	83.85	56.66	−0.39	0.7	0.06

Abbreviations: NT-pro-BNP–baseline—preoperative Nt-pro-BNP serum concentration; NT-pro-BNP–24 h—Nt-pro-BNP serum concentration 24 h after the surgery; NT-pro-BNP–48 h—Nt-pro-BNP serum concentration 48 h after the surgery; Δ NT-pro-BNP–1-0—Difference in NT-proBNP concentration between the first postoperative day (NT-pro-BNP–24 h) and the preoperative Nt-pro-BNP measurement; Δ NT-pro-BNP–2-0—Difference in NT-proBNP concentration between the second postoperative day (NT-pro-BNP–48 h) and the preoperative Nt-pro-BNP measurement; %Δ NT-pro-BNP–1-0—Percentage change in NT-proBNP concentration between the first postoperative day (NT-pro-BNP–24 h) and the preoperative Nt-pro-BNP measurement; %Δ NT-pro-BNP–2-0—Percentage change in NT-proBNP concentration between the second postoperative day (NT-pro-BNP–48 h) and the preoperative Nt-pro-BNP measurement; %Δ NT-pro-BNP–2-1—Percentage change in NT-proBNP concentration between the second postoperative day (NT-pro-BNP–48 h) and the first postoperative day (NT-pro-BNP–24 h); IQR—interquartile range; Min.—minimum value; Max.—maximum value.

## Data Availability

Data are available upon request from the corresponding author.

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
