# Peer review of "Individualized Perioperative Hemodynamic Management Using Hypotension Prediction Index Software and the Dynamics of Troponin and NTproBNP Concentration Changes in Patients Undergoing Oncological Abdominal Surgery"

_jpm, 2024, doi:10.3390/jpm14020211_

Round 1

Reviewer 1 Report

Comments and Suggestions for Authors

Paper represents well designed single center investigation with relatively small number of participants.

I have few important comments to adress:

Authors investigated usage of hypotension prediction index in managing anesthesia for high risk cardiac pts in non cardiac surgery. In this moment this technology is not new, and there are several papers that already investigated possible advantages of using it. However, there is a lack of large randomised studies in this field that will give us more answers of impact of HPI on postoperative adverse events. Small single-center studies like this one was already published previously, supporting conlusion like the one from this paper.

Authors did not explained enough to be understud advantage of prediction of hypotension in patient monitoring in Introduction section, nor did not mentioned any of this published materials also in Introduction  section, (for example study of Šribar A et al, with similar results that can be compared). It is important to expand the Introduction with few of this studies to give explanation for that what is already known about HPI. 

In addition to it, In Perioperative Management section authors mentioned among other usual parameters of cardiac monitoring two specific parameters for this monitoring technology: dP/dt – contractility, and Eadyn - dynamic vascular elastance.

Nevertheless, authors further stated that  ..." This study did not adhere to a predefined specific treatment protocol for the included patients; rather, patient management was adapted according to the clinical routine of the attending anesthesiologist. Throughout the anesthesia, individualized cardiovascular function optimization was conducted based on data from the monitoring platform."... 

The important idea and possible advantage of this advanced monitoring is to go along with your treatment decision upon parameter changes; first HPI and then on dynamics of other parameters on which way you will manage hypotension.

So, it is not precisely said did the anesthesiologists in charge react upon parameters changes or just upon HPI to signal hypotension and then manage it without further analysis of monitor parameters?

That is why you need to explaine importance of Eadyn and dP/dt, and better define when you used fluid, inotrope or pressors. From what you state it is not clear when you will decide to use one of this treatment options.

Please define it and explaine together with Eadyn and dP/dt.

Author Response

Review 1 :

Reviewer 1: Paper represents well designed single center investigation with relatively small number of participants. I have few important comments to adress:

Authors: Thank you for dedicating your time and expertise to review our manuscript.

Reviewer 1: Authors investigated usage of hypotension prediction index in managing anesthesia for high risk cardiac pts in non cardiac surgery. In this moment this technology is not new, and there are several papers that already investigated possible advantages of using it. However, there is a lack of large randomised studies in this field that will give us more answers of impact of HPI on postoperative adverse events. Small single-center studies like this one was already published previously, supporting conlusion like the one from this paper.

Authors: We appreciate your insightful comment. Indeed, previous studies have investigated the application of the Hemodynamic Performance Index (HPI) in general anesthesia and its effectiveness in mitigating episodes of hypotension during these procedures. However, our study uniquely focuses on two specific aspects: firstly, evaluating the efficacy of HPI in a high-risk oncologic population, and secondly, analyzing the dynamics of cardiac biomarkers, namely NT-pro-BNP and troponin, in this demographic, particularly considering the high proportion of patients with preoperatively diagnosed heart failure. Our extensive literature review, conducted via PubMed and Medline databases, suggests that this specific area has not been comprehensively studied before. Significantly, research has demonstrated a correlation between post-operative NT-pro-BNP concentrations and Major Adverse Cardiac Events (MACE). For example, Rodseth et al. (2014) highlighted the prognostic value of pre-operative and post-operative B-type natriuretic peptides in patients undergoing noncardiac surgery, as detailed in their systematic review and individual patient data meta-analysis (J Am Coll Cardiol. 2014;63(2):170-180. doi:10.1016/j.jacc.2013.08.1630). These findings are discussed in our discussion section. Our primary aim was to investigate whether controlled, individualized hemodynamic monitoring using HPI could reduce hemodynamic stress, as expressed by NT-pro-BNP and troponin levels, in populations both with and without heart failure.

Reviewer 1: Authors did not explained enough to be understud advantage of prediction of hypotension in patient monitoring in Introduction section, nor did not mentioned any of this published materials also in Introduction  section, (for example study of Šribar A et al, with similar results that can be compared). It is important to expand the Introduction with few of this studies to give explanation for that what is already known about HPI. 

Authors: We appreciate your valuable feedback. In accordance with your recommendation, we have expanded the introduction section of our manuscript to encompass a more detailed review of previously published studies involving the Hemodynamic Performance Index (HPI). These additions have been clearly marked with a yellow highlight for ease of identification and review.

Reviewer 1: In addition to it, In Perioperative Management section authors mentioned among other usual parameters of cardiac monitoring two specific parameters for this monitoring technology: dP/dt – contractility, and Eadyn - dynamic vascular elastance.

Nevertheless, authors further stated that  ..." This study did not adhere to a predefined specific treatment protocol for the included patients; rather, patient management was adapted according to the clinical routine of the attending anesthesiologist. Throughout the anesthesia, individualized cardiovascular function optimization was conducted based on data from the monitoring platform."... 

The important idea and possible advantage of this advanced monitoring is to go along with your treatment decision upon parameter changes; first HPI and then on dynamics of other parameters on which way you will manage hypotension.

So, it is not precisely said did the anesthesiologists in charge react upon parameters changes or just upon HPI to signal hypotension and then manage it without further analysis of monitor parameters?

That is why you need to explaine importance of Eadyn and dP/dt, and better define when you used fluid, inotrope or pressors. From what you state it is not clear when you will decide to use one of this treatment options.

Please define it and explaine together with Eadyn and dP/dt.

Authors: We appreciate your observation. In our study, the anesthesiologist overseeing each case had full access to the array of hemodynamic parameters displayed by the Edwards Lifesciences console, which included the Hemodynamic Performance Index (HPI). Decisions regarding patient management were made based on a combination of these parameters - blood pressure, cardiac output, stroke volume, stroke volume variation, pulse pressure variation, dP/dt, and dynamic vascular elastance - and not on any single parameter independently. These decisions, encompassing fluid administration, vasopressors, or inotropes, were informed by the clinician’s expertise and specific training in HPI, rather than a predefined algorithm tailored to this study. The HPI platform served as a tool to aid the anesthesiologists, who applied it in conjunction with their clinical acumen and training. We have restructured the sentence in the manuscript to better clarify this approach. For better understandment of clinical definitions used in advanced hemodynamic monitoring we added those definitions to Clinical Definitions section. Additionally, we have provided a new figure, a photograph of the screen of the Edwards Lifescience console with the HPI module (new Figure 1 ). All changes are marked with yellow highlight.

Reviewer 2 Report

Comments and Suggestions for Authors

This is an interesting prospective study involving patients scheduled for major cancer gastrointestinal surgery investigating the usefulness of perioperative hemodynamic monitoring. However, some points need to be addressed before publication.

1]Methods: Hemodynamic monitoring is an essential part of this work. Therefore, a separate section should be added describing the basic measuring principle of the Acumen sensor (with a photo), the definitions of HPI, CI, SV, SVRI, SVV, dP/dt, and Eadyn and how they are approximated by the Edwards Lifesciences software.

The corresponding parts of the sections “Study Design” and “Clinical Definitions” should move into this new section.

2] Results, Table 1: The age range of the patient cohort is very large and the NT-proBNP physiological concentration depends on age and sex. How did the authors cope with this?

3] Results, Page 9: The significance of the correlation between Troponin concentration and number of HPI alarms, as well as between NT-proBNP concentration and number of HPI alarms, should be examined both before and after the surgery.

Author Response

Reviewer 2: This is an interesting prospective study involving patients scheduled for major cancer gastrointestinal surgery investigating the usefulness of perioperative hemodynamic monitoring. However, some points need to be addressed before publication.

Authors: Thank you for agreeing to review my submission. Your time and expertise are much appreciated, and I am grateful for your willingness to provide feedback.

Reviewer 2: (1) Methods: Hemodynamic monitoring is an essential part of this work. Therefore, a separate section should be added describing the basic measuring principle of the Acumen sensor (with a photo), the definitions of HPI, CI, SV, SVRI, SVV, dP/dt, and Eadyn and how they are approximated by the Edwards Lifesciences software. The corresponding parts of the sections “Study Design” and “Clinical Definitions” should move into this new section.

Authors: We are grateful for your insightful suggestion. Following your advice, we have augmented the 'Clinical Definitions' section of our manuscript. This section now includes a comprehensive elaboration of advanced hemodynamic parameters employed in conjunction with the Hemodynamic Performance Index (HPI). We have distinctly marked these modifications with a yellow highlight for ease of review. We kindly request your assessment of these changes to determine if they sufficiently address your concerns or if further revisions are necessary.

Reviewer 2 (2) Results, Table 1: The age range of the patient cohort is very large and the NT-proBNP physiological concentration depends on age and sex. How did the authors cope with this?

Thank you for this insightful comment. Our approach to the wide age range and sex-related differences in NT-proBNP physiological concentrations involved focusing on the absolute values and changes in NT-proBNP concentration, rather than categorizing them as 'normal' or 'above normal.' This decision was based on the understanding that NT-proBNP levels vary significantly with age and sex.

Reviewer 2 – (3) Results, Page 9: The significance of the correlation between Troponin concentration and number of HPI alarms, as well as between NT-proBNP concentration and number of HPI alarms, should be examined both before and after the surgery.

We acknowledge the reviewer's suggestion regarding the significance of correlating Troponin and NT-proBNP concentrations with the number of HPI alarms in both the preoperative and postoperative periods. In our study, patients were selected based on their clinical stability and did not require catecholamine amines preoperatively, resulting in uniformly low pre-induction HPI values. Advanced hemodynamic monitoring was concluded at the end of the surgical procedure. We recognize that the risk of hypotension can be significant in the postoperative period. However, the study design intentionally excluded the postoperative assessment. This decision was based on the practical challenges of continuous monitoring during the initial postoperative phase, where patients often undergo rehabilitation and mobilization. Future studies could benefit from extending the monitoring into the postoperative period to explore these correlations further.

Round 2

Reviewer 2 Report

Comments and Suggestions for Authors

The manuscript is now improved however, there are still some points that need attention:

1]Methods: Hemodynamic monitoring is an essential part of this work so, the basic measuring principle and technology of the Acumen IQ sensor should be described. Since one of the authors is affiliated with Edwards Lifesciences, this should not be too hard. Is it an invasive or non-invasive sensor?

2] Methods: In addition to the standard definitions of the HPI, CI, SV, SVRI, SVV, dP/dt, and Eadyn hemodynamic parameters, a brief description of how they are estimated from the shape and characteristics of the arterial waveform should be given. This will give a general idea of how the HemoSphere platform and the HPI software work.

3] Methods: The photos in Figure 1 are not clear.

Author Response

Reviewer: Methods: Hemodynamic monitoring is an essential part of this work so, the basic measuring principle and technology of the Acumen IQ sensor should be described. Since one of the authors is affiliated with Edwards Lifesciences, this should not be too hard. Is it an invasive or non-invasive sensor?

Authors: Thank you for this point. We have added the information about minimally invasive access through the radial artery line used in this study. We have added this information in the updated methods section. All changes are marked with yellow highlights. 

Reviewer: 2] Methods: In addition to the standard definitions of the HPI, CI, SV, SVRI, SVV, dP/dt, and Eadyn hemodynamic parameters, a brief description of how they are estimated from the shape and characteristics of the arterial waveform should be given. This will give a general idea of how the HemoSphere platform and the HPI software work.

Authors: Thank you for highlighting this aspect. In response, we have updated the whole methods section to outline the basics of the hemodynamic monitoring technology developed by Edwards Lifesciences. Specifically, we address the measurement technique (arterial waveform analysis) and acquisition of the derived parameters by the Acumen IQ sensor. We believe that this inclusion provides a clearer understanding of the technological and analytical aspects of this hemodynamic monitoring approach.

Reviewer: 3] Methods: The photos in Figure 1 are not clear.

Thank you for this comment. To make the reception of  Figure 1 more clear we have improved the description of the photos in this figure. We hope this adds to the clarity of the picture's educational aspect.